# The association between clinical, subclinical features and autoantibody profile in Vietnamese dermatomyositis patients

**My Huyen Le**[1]*, **Hoa Thi Dinh**[1], **Thu Thi Hoai Le**[1], **Phuong Thi Hoang**[1], **Minh Nguyet Vu**[1,2], **Doanh Huu Le**[1,2]

**1** National Hospital of Dermatology & Venereology, Hanoi, Vietnam, **2** Hanoi Medical University, Hanoi, Vietnam

* drlehuyenmy@gmail.com

**Data Availability Statement:** All data are in the manuscript and Supporting information files.

**Funding:** The authors received no specific funding for this work.

## Abstract

There is still a lack of research in Vietnam on the autoantibody profile of dermatomyositis (DM) and its association with clinical and subclinical characteristics. Therefore, we conducted this study to investigate clinical and subclinical correlations with autoantibodies in DM patients. 72 DM patients at Vietnam National Hospital of Dermatology and Venereology (NHDV) from March 2019 to September 2021 were included in this cross-sectional study. Clinical manifestations and laboratory test results of the patients were obtained at the time of visit. Of these, 63 patients were tested for the presence of autoantibodies using an Immunoblot assay. Our findings show that the average age of patients was 41.7 years. The female-male ratio was 1.7:1. The most common skin and muscle manifestations were myalgia (79.2%), heliotrope rash (62.5%), shawl sign (61.1%), Gottron's sign (59.7%), muscle weakness (59.7%), Gottron's papule (52.8%), periungual telangiectasia (41.7%), V-sign (38.9%), poikiloderma (26.4%), periungual fissures (20.8%), Raynaud's phenomenon (15.3%). Among the 63 patients tested for autoantibodies, myositis-specific antibodies (MSAs) were found in 71.4% of the serum samples, and myositis-associated antibodies (MAAs) in 36.5%. Anti-TIF1γ antibody accounted for the highest percentage (28.6%), followed by anti-Ro52 (22.2%), anti-synthetase (17.5%), anti-Mi-2 and anti-MDA5 (both 14.3%). Anti-synthetase antibodies (ARS-Abs) showed a significant association with arthralgia, fever, and Raynaud's phenomenon, while anti-TIF1γ antibodies showed a strong association with V-sign and poikiloderma (p<0.05). Clinical features in dermatomyositis are heterogeneous. Our study results show some associations between clinical features and autoantibodies in patients with DM. The analysis of DM-related autoantibodies is clinically useful, will be essential for the approaches to diagnosis, and management of DM patients.

## Introduction

Dermatomyositis (DM) represents a group of chronic inflammatory disorders characterized by myogenic changes and/or skin eruptions. Clinical features are heterogeneous with

**Competing interests:** The authors have declared that no competing interests exist.

predominant proximal muscle weakness, typical skin rashes such as Gottron's papules/sign, heliotrope erythema, V sign (sunburn reaction over the 'V' area of the anterior chest), nail fold changes (periungual fissures, periungual telangiectasia, periungual hemorrhage). Apart from skin and muscle damage, patients may also have symptoms with internal organs: joints, lungs, heart, and digestive tract, which can greatly affect their daily activities, and sometimes could be life-threatening.

The presence of diagnostic autoantibodies (Abs), known as myositis-related Abs, is a prominent feature. Several studies have reported the association between DM autoantibodies and clinical manifestations of the disease [1, 2]. Anti-Jo-1 is one of the anti-synthetase antibodies (ARS-Abs), which has an important value in the diagnosis of DM. It is also a prognostic factor with over 70% of positive anti-Jo-1 DM patients have developed interstitial lung disease (ILD) [3]. These patients often have a more severe clinical picture, a higher frequency of recurrence, and a worse prognosis because they often have a poor response to treatment. Some other ARS-Abs include anti-PL-7, anti-PL-12, anti-EJ and anti-OJ. Anti-PL12 is common in DM patients without muscle damage or with ILD alone. ILD was diagnosed in 92–100% in the anti-PL-12-positive group compared with 50–75% in the anti-Jo-1-positive group [4]. In contrast to anti-Jo-1, positive anti-Mi-2 is often associated with typical skin lesions but not severe muscle damage and often responds well to treatment. Therefore, DM patients with anti-Mi-2 antibodies have a better prognosis [2]. TIF-1 (transcription intermediary factor 1) is a transcriptional mediator that plays an important role in cellular functions, especially mutations that lead to cancer. Anti-TIF-1γ antibodies act against proteins produced in the immune response against cancer and thus increase the risk of cancer in DM patients with positive anti-TIF-1γ. The proportion of this patients associated with cancer ranges from 42 to 75% [5]. Anti-MDA5 is common in atypical DM patients with little or no muscle damage. Sato et al. conducted a study of anti-MDA5 antibodies and determined that these antibody levels may be related to disease activity, as they decrease and become negative in treatment responsive patients [6]. Thus, anti-MDA5 antibody levels can be used as a biomarker of disease activity and a predictor of response to treatment. The presence of anti-Ro52 antibodies has been reported in a variety of autoimmune diseases, especially myositis, systemic sclerosis and autoimmune liver diseases. However, the association of antibodies against Ro52/TRIM21 with clinical manifestations remains controversial. As previously reported, anti-Ro52 is the most common autoantibody detected in myositis with anti-synthetase syndrome. Several studies reported an association between anti-Ro52 antibody and anti-tRNA synthetases syndrome [7].

In Vietnam, there is an absence of a fully developed enzyme-linked immunosorbent assay (ELISA)-based system for analyzing myositis-related Abs. Recently, the use of EUROLINE strips (based on the immunoblot method) allows us to detect multiple DM-specific Abs and DM-associated Abs simultaneously. Therefore, this study aims to investigate the correlation between clinical, subclinical manifestations, and autoantibodies profile in Vietnamese patients with DM.

## Materials and methods

### Subjects and study design

This cross-sectional study included 72 patients with DM who were treated at Vietnam National Hospital of Dermatology and Venereology (NHDV) from March 2019 to September 2021, both retrospectively and prospectively. The diagnosis of DM was based on the classification criteria of Tanimoto et al. in 1995 and overlap syndromes were excluded [3]. Patients were clinically examined on clinical manifestations, and indicated subclinical procedures such as skin biopsy, muscle biopsy, electromyography and blood biochemistry tests. Laboratory

tests were considered abnormal above the following values for creatin kinase (CK), aspartate aminotransferase (AST) and alanine aminotransferase (ALT), respectively: 190 U/L, 40 U/L and 40 U/L. All patients were screened for malignancy, including clinical examination, abdominal ultrasound, chest X-ray. If there were suspicious symptoms, the patient would undergo gastroscopy, high resolution computed tomography (HRCT) where applicable.

EUROLINE strips (by EUROIMMUN Medizinische Labordiagnostika AG) based on the immunoblot method which uses membrane strips coated with several purified, biochemically characterized antigens at specific positions which allow the detection of multiple autoantibodies simultaneously. The assay is fast in interpreting results compared to immunoprecipitation. Comprehensive studies performed in major European centers have shown that using the immunoblot technique to identify autoantibodies is more valuable, specific, and useful for subtypes of IIMs in general and for DM patients in particular [1].

The incubation protocol includes the 5 steps: (1) Pretreat—put the test strip into the incubation channel and fill each channel with 1.5ml sample buffer, incubate for 5 minutes at room temperature (+18°C to +25°C) on a rocking shaker. (2) 1st incubate—aspirate off, pipette 1.5 ml of diluted serum sample (1:101) into the incubation channel, incubate at room temperature for 30 minutes on a rocking shaker then aspirate off, wash 3x5 min with 1.5 ml working strength wash buffer each. (3) 2nd incubate—aspirate off, pipette 1.5 ml enzyme conjugate into the incubation channel, incubate for 30 minutes at room temperature on a rocking shaker then aspirate off, wash 3x5 min with 1.5 ml working strength wash buffer each. (4) 3rd incubate—aspirate off, pipette 1.5 ml substrate into the incubation channel, incubate for 10 minutes at room temperature on a rocking shaker then aspirate off, rinse three times with 1.5 ml distilled water. (5) Evaluation—place test strip on the evaluation protocol (EUROLineScan), air dry, and evaluate.

## Ethical issues

The study was approved by the Institutional Review Board of Hanoi Medical University (approval number: 2396/QD-DHYHN) and adhered to the principles of the Declaration of Helsinki. Written informed consent was obtained from all participant and the parent/guardian of participants under 18 years of age. Specimens used in this study are part of routine patient management without any additional samples drawn. There was no objection for their samples to be used.

## Statistical methods

Data were both entered and analyzed by SPSS 20.0 software (version 20.0; SPSS Inc., Chicago, IL, USA). The results are shown as means (standard deviation), median (range), or proportions. For numeric data, Pearson's $\chi^2$ test or Fisher's exact test was used, and $p < 0.05$ was considered significant.

## Results

### The baseline characteristics of 72 DM patients

Of 72 subjects, 55 patients were adult DM, and 17 were juvenile DM (Table 1). The female-male ratio was 1.7:1. There was no statistically significant difference between the gender groups for adult DM and juvenile DM (p = 0.431). The age group from 41 to 60 was the most, accounting for 29.2%. The average age of all subjects was 41.7 years. The average age of disease onset of the juvenile DM group and adult DM group were 8.0 years and 49.3 years, respectively.

**Table 1. The baseline characteristics of research subjects (n = 72).**

| General characteristics | | Adult dermatomyositis | Juvenile dermatomyositis |
|---|---|---|---|
| Average age of onset (years old) | | 49.3 ± 17.7 | 8.0 ± 3.3 |
| Gender | Male | 22 | 5 |
| | Female | 33 | 12 |
| | p-value | 0.431[a] | |
| Age group (%) | 0–9 | 5.6 | |
| | 10–17 | 18.1 | |
| | 18–40 | 20.8 | |
| | 41–60 | 29.2 | |
| | 61–80 | 25.0 | |
| | Over 80 | 1.4 | |
| Age (years old) | Mean ± sd | 41.7 ± 22.9 | |
| | Min | 3 | |
| | Max | 91 | |

[a]: Pearson's χ2 test (compare two groups, adult DM and juvenile DM)

## Clinical and subclinical characteristics of the study subjects

The most common skin and muscle manifestations found in our study were myalgia (79.2%), heliotrope rash (62.5%), shawl sign (61.1%), Gottron's sign (59.7%), muscle weakness (59.7%), Gottron's papule (52.8%), periungual telangiectasia (41.7%), V-sign (38.9%), poikiloderma (26.4%), periungual fissures (20.8%), Raynaud's phenomenon (15.3%) (Table 2).

The shawl sign appeared in the adult DM group (70.9%) more frequently than the juvenile DM group (29.4%), the difference was significant statistically with p<0.05. There was no statistical difference in elevated CK, AST and ALT enzymes between the two observed groups.

**Systemic and organ symptoms.** In total, fatigue was the most common symptom (58.3%), followed by weight loss (30.6%), hair loss (27.8%), arthralgia (27.8%), exertional dyspnea (18.1%), heartburn (16.7%), lymphadenopathy (13.9%), fever (12.5%), and itch 5.8%. Hypertension was only seen in the adult DM group (5.6%). There was no difference between adult DM and juvenile DM regarding systemic symptoms and organ damage (Table 3).

A total of 63 serum samples from patients with DM were tested for autoantibody profile (Table 4). Of these samples, 50 (79.4%) were reactive with at least one autoantibody. Myositis-specific antibodies (MSAs) were found in 71.4% of the serum samples, and myositis-associated antibodies (MAAs) in 36.5%. Anti-TIF1γ antibody accounted for the highest percentage (28.6%), followed by anti-Ro52 (22.2%), anti-synthetase (17.5%), anti-Mi-2 and anti-MDA5 (both 14.3%).

Anti-synthetase antibodies showed significant association with arthralgia, fever and Raynaud's phenomenon, while anti-TIF1γ antibody showed a strong association with V-sign and poikiloderma (p<0.05). Clinical and subclinical manifestations between positive and negative anti-Ro52 or anti-Mi-2 or anti-MDA5 antibody (p>0.05) were not significant statistically (Table 5).

## Discussion

We investigate clinical features and myositis-related antibodies of 72 DM patients. All DM patients included in our study had significant characteristic skin manifestations. Skin lesions of DM are diverse and mandatory diagnostic criteria. Heliotrope rash is an important

**Table 2. Skin and muscle manifestations (n = 72).**

| Manifestations | n(%) | | | p-value |
|---|---|---|---|---|
| | Adult dermatomyositis (n = 55) | Juvenile dermatomyositis (n = 17) | Total (n = 72) | |
| • Heliotrope rash | 34 (61.8) | 11 (64.7) | 45 (62.5) | 0.83[a] |
| • V-sign | 21 (38.2) | 7 (41.2) | 28 (38.9) | 0.825[a] |
| • *Shawl sign* | *39 (70.9)* | *5 (29.4)* | *44 (61.1)* | *0.002*[a] |
| • Gottron's papule | 28 (50.9) | 10 (58.8) | 38 (52.8) | 0.568[a] |
| • Gottron's sign | 32 (58.2) | 11 (64.7) | 43 (59.7) | 0.632[a] |
| • Poikiloderma | 14 (25.5) | 5 (29.4) | 19 (26.4) | 0.759[b] |
| • Calcinosis cutis | 2 (3.6) | 1 (5.9) | 3 (4.2) | 0.56[b] |
| • Mechanic's hands | 7 (12.7) | 0 | 7 (9.7) | 0.187[b] |
| • Periungual fissures | 14 (25.5) | 1 (5.9) | 15 (20.8) | 0.1[b] |
| • Periungual telangiectasia | 23 (41.8) | 7 (41.2) | 30 (41.7) | 0.963[a] |
| • Periungual hemorrhage | 6 (10.9) | 0 | 6 (8.3) | 0.325[b] |
| • Raynaud's phenomenon | 8 (14.5) | 3 (17.6) | 11 (15.3) | 0.714[b] |
| • Skin ulcers | 5 (9.1) | 1 (5.9) | 6 (8.3) | 1.0[b] |
| • Myalgia | 43 (78.2) | 14 (82.4) | 57 (79.2) | 1.0[b] |
| • Muscle weakness | 33 (60) | 10 (58.8) | 43 (59.7) | 0.931[a] |
| • Muscle atrophy | 6 (10.9) | 0 | 6 (8.3) | 0.325[b] |
| • Clinical muscle damages (at least one of these symptoms: myalgia, muscle weakness, muscle atrophy) | 47 (75.8) | 15 (24.2) | 62 (86.1) | 1.0[b] |
| **Serum muscle enzymes** | **Clinical muscle damage (n = 62)** | **No clinical muscle damage (n = 10)** | **Total (n = 72)** | |
| • Elevated CK enzyme | 25 (40.3) | 3 (30) | 28 (38.9) | 0.73[b] |
| • Elevated AST enzyme | 26 (41.9) | 4 (40) | 30 (41.7) | 1.0[b] |
| • Elevated ALT enzyme | 22 (35.5) | 3 (30) | 25 (34.7) | 1.0[b] |

[a]: Pearson's χ2 test,

[b]: Fisher's Exact test (compare two groups, adult DM and juvenile DM)

symptom to differentiate DM from other diseases. In our study, the rate of Heliotrope rash was 62.5%, this is not particularly high compared to previous rates reported by Thuy Nguyen Thi Phuong et al. and Dourmishev et al. (76% and 88.1%, respectively) [8, 9]. Meanwhile, Gottron's papules and Gottron's sign were seen in 59.7% and 52.8%, respectively. There was no statistical significant difference between adult DM and juvenile DM for the different skin lesions, except for the shawl sign (61.1%), which was observed more in adult DM (70.9%) as compared to juvenile DM (29.4%) with p<0.05 (Table 2).

Vascular lesions in DM include Raynaud's phenomenon, skin ulcers, periungual telangiectasia, or hemorrhage. Periungual telangiectasia is also a common symptom in our study (41.7%), Thuy Nguyen Thi Phuong et al. found a similar result (42.9%) [8], but a higher percentage was reported by Gowdie et al. (68%) [10]. Periungual hemorrhage is the later stage of periungual telangiectasia, but only presented in 8.3% of patients. Calcinosis cutis appears with variable rates among studies and are often nonspecific for DM, which can be seen in other autoimmune diseases. A previous study recorded calcinosis cutis was 74% in the juvenile DM group, and 20% in the adult DM group [11], however, in our study, this lesion in both groups was very low (5.9% and 3.6%, respectively).

Myalgia is the most common symptom of classical DM and one of the diagnostic criteria. We found that clinical muscle damages (including myalgia or muscle weakness or muscle

**Table 3. Systemic and organ symptoms (n = 72).**

| Characteristics | n (%) | | | p-value |
|---|---|---|---|---|
| | Adult dermatomyositis (n = 55) | Juvenile dermatomyositis (n = 17) | Total (n = 72) | |
| • Fever | 8 (14.5) | 1 (5.9) | 9 (12.5) | 0.676[b] |
| • Fatigue | 35 (63.6) | 7 (41.2) | 42 (58.3) | 0.101[a] |
| • Hair loss | 17 (30.9) | 3 (17.6) | 20 (27.8) | 0.364[b] |
| • Lose weight | 18 (32.7) | 4 (23.5) | 22 (30.6) | 0.472[a] |
| • Lymphadenopathy | 7 (12.7) | 3 (17.6) | 10 (13.9) | 0.691[b] |
| • Itchy | 3 (5.5) | 1 (5.9) | 4 (5.6) | 1.0[b] |
| • Hypertension | 4 (7.3) | 0 | 4 (5.6) | 0.566[b] |
| • Exertional dyspnea | 12 (21.8) | 1 (5.9) | 13 (18.1) | 0.17[b] |
| • Dry cough | 3 (5.5) | 0 | 3 (4.2) | 1.0[b] |
| • Dysphagia | 5 (9.1) | 1 (5.9) | 6 (8.3) | 1.0[b] |
| • Heartburn | 10 (18.2) | 2 (11.8) | 12 (16.7) | 0.719[b] |
| • Diarrhea | 1 (1.8) | 0 | 1 (1.4) | 1.0[b] |
| • Constipation | 2 (3.6) | 0 | 2 (2.8) | 1.0[b] |
| • Arthralgia | 17 (30.9) | 3 (17.6) | 20 (27.8) | 0.364[b] |
| • Cardiac arrhythmias | 3 (5.5) | 0 | 3 (4.2) | 1.0[b] |

[a]: Pearson's χ2 test,

[b]: Fisher's Exact test (compare two groups, adult DM and juvenile DM)

**Table 4. The profile of autoantibodies of 63 tested patients (n = 63).**

| Autoantibodies | n (%) | | | p-value |
|---|---|---|---|---|
| | Adult dermatomyositis (n = 49) | Juvenile dermatomyositis (n = 14) | Total (n = 63) | |
| • Negative | 9 (18.4) | 4 (28.6) | 13 (20.6) | 0.461[b] |
| • 1 positive autoantibody | 21 (42.9) | 4 (28.6) | 25 (39.7) | 0.355[a] |
| • 2 positive autoantibodies | 11 (22.4) | 3 (21.4) | 14 (22.2) | 1.0[b] |
| • ≥3 positive autoantibodies | 8 (16.3) | 3 (21.4) | 11 (17.5) | 0.696[b] |
| • *Myositis-specific autoantibodies* | 36 (73.5) | 9 (64.3) | 45 (71.4) | 0.517[b] |
| Anti-synthetase | 8 (16.3) | 3 (21.4) | 11 (17.5) | 0.696[b] |
| Anti-Jo-1 | 3 (6.1) | 2 (14.3) | 5 (7.9) | 0.307[b] |
| Anti-PL-7 | 2 (4.1) | 2 (14.3) | 4 (6.3) | 0.211[b] |
| Anti-PL-12 | 3 (6.1) | 0 | 3 (4.8) | 1.0[b] |
| Anti-EJ | 1 (2.0) | 0 | 1 (1.6) | 1.0[b] |
| Anti-TIF1γ | 13 (26.5) | 5 (35.7) | 18 (28.6) | 0.517[b] |
| Anti-Mi-2 | 7 (14.3) | 2 (14.3) | 9 (14.3) | 1.0[b] |
| Anti-MDA5 | 8 (16.3) | 1 (7.1) | 9 (14.3) | 0.67[b] |
| Anti-SAE1 | 5 (10.2) | 0 | 5 (7.9) | 0.578[b] |
| Anti-NXP2 | 2 (4.1) | 3 (21.4) | 5 (7.9) | 0.068[b] |
| Anti-SRP | 3 (6.1) | 2 (14.3) | 5 (7.9) | 0.307[b] |
| • *Myositis-associated autoantibodies* | 19 (38.8) | 4 (28.6) | 23 (36.5) | 0.484[a] |
| Anti-Ro52 | 13 (26.5) | 1 (7.1) | 14 (22.2) | 0.162[b] |
| Anti-Ku | 6 (12.2) | 2 (14.3) | 8 (12.7) | 1.0[b] |
| Anti-PM/Scl | 3 (6.1) | 2 (14.3) | 5 (7.9) | 0.307[b] |

[a]: Pearson's χ2 test,

[b]: Fisher's Exact test (compare two groups, adult DM and juvenile DM)

**Table 5. Associations of autoantibodies with symptoms.**

| Manifestations | Positive anti-synthetase (n = 11) | | Positive anti-TIF1γ (n = 18) | | Positive anti-Ro52 (n = 14) | | Positive anti-Mi-2 (n = 9) | | Positive anti-MDA5 (n = 9) | |
|---|---|---|---|---|---|---|---|---|---|---|
| | n (%) | p-value | n (%) | p-value | n (%) | p-value | n (%) | p-value | n (%) | p-value |
| Heliotrope rash | 7 (63.6) | 1.0[b] | 13 (72.2) | 0.363[a] | 10 (71.4) | 0.484[a] | 7 (77.8) | 0.467[b] | 8 (88.9) | 0.137[b] |
| V-sign | 4 (36.4) | 1.0[a] | **11 (61.1)** | **0.028[a]** | 4 (28.6) | 0.335[a] | 6 (66.7) | 0.138[b] | 2 (22.2) | 0.298[b] |
| Shawl sign | 7 (63.3) | 1.0[b] | 13 (72.2) | 0.554[a] | 10 (71.4) | 0.757[b] | 6 (66.7) | 1.0[b] | 4 (44.4) | 0.146[b] |
| Gottron's papule | 6 (54.5) | 1.0[b] | 9 (50) | 0.811[a] | 9 (64.3) | 0.312[a] | 4 (44.4) | 0.725[b] | 2 (22.2) | 0.073[b] |
| Gottron's sign | 7 (63.6) | 1.0[b] | 14 (77.8) | 0.101[a] | 8 (57.1) | 0.677[a] | 5 (55.6) | 0.721[b] | 3 (33.3) | 0.073[b] |
| Poikiloderma | 5 (45.5) | 0.149[b] | **9 (50)** | **0.014[b]** | 6 (42.9) | 0.174[b] | 2 (22.2) | 1.0[b] | 0 | 0.098[b] |
| Calcinosis cutis | 2 (18.2) | 0.076[b] | 1 (5.6) | 1.0[b] | 0 | 1.0[b] | 0 | 1.0[b] | 1 (11.1) | 0.375[b] |
| Mechanic's hands | 3 (27.3) | 0.095[b] | 3 (16.7) | 0.397[b] | 3 (21.4) | 0.177[b] | 0 | 0.58[b] | 1 (11.1) | 1.0[b] |
| Periungual fissures | 2 (18.2) | 1.0[b] | 7 (38.9) | 0.104[b] | 5 (35.7) | 0.291[b] | 1 (11.1) | 0.673[b] | 1 (11.1) | 0.673[b] |
| Periungual hemorrhage | 0 | 0.579[b] | 4 (22.2) | 0.051[b] | 2 (14.3) | 0.607[b] | 1 (11.1) | 1.0[b] | 1 (11.1) | 1.0[b] |
| Periungual telangiectasia | 5 (45.5) | 0.75[b] | 9 (50) | 0.373[a] | 6 (42.9) | 0.891[a] | 4 (44.4) | 1.0[b] | 3 (33.3) | 0.725[b] |
| Raynaud's phenomenon | **6 (54.5)** | **0.000[b]** | 3 (16.7) | 0.707[b] | 1 (7.1) | 0.67[b] | 2 (22.2) | 0.604[b] | 1 (11.1) | 1.0[b] |
| Exertional dyspnea | 0 | 0.187[b] | 4 (22.2) | 0.714[b] | 4 (28.6) | 0.243[b] | 3 (33.3) | 0.184[b] | 3 (33.3) | 0.184[b] |
| Fever | **6 (54.5)** | **0.000[b]** | 2 (11.1) | 1.0[b] | 2 (14.3) | 1.0[b] | 1 (11.1) | 1.0[b] | 1 (11.1) | 1.0[b] |
| Arthralgia | **7 (63.6)** | **0.004[b]** | 4 (22.2) | 1.0[b] | 3 (21.4) | 1.0[b] | 2 (22.2) | 1.0[b] | 4 (44.4) | 0.214[b] |
| Clinical muscle damage | 10 (90.9) | 0.676[b] | 14 (77.8) | 0.452[b] | 11 (78.6) | 0.679[b] | 9 (100) | 0.332[b] | 9 (100) | 0.332[b] |
| Elevated CK enzyme | 5 (45.5) | 0.75[b] | 9 (50) | 0.373[a] | 7 (50) | 0.452[a] | 3 (33.3) | 0.725[b] | 3 (33.3) | 0.725[b] |
| Malignancy | 0 | . | **1 (5.6)** | **0.286[b]** | 0 | . | 0 | . | 0 | . |

[a]: Pearson's χ2 test,

[b]: Fisher's Exact test (compare two groups, negative and positive autoantibodies)

atrophy) were 86.1%, of which, myalgia was the most common (79.2%). Muscle weakness is a specific symptom of DM. Among 72 patients, 43 patients (59.7%) had muscle weakness. A similar result was obtained by Bohan et al. (69%) [12], but a higher percentage was reported by Thuy Nguyen Thi Phuong et al. (97%) [8]. This could be due to an inpatient cohort was recruited for the study where most cases were severe with aggressive clinical symptoms while this study was based on an outpatient cohort where most were already on stable treatment.

In DM, when the muscle cells are damaged by inflammation and necrosis, they release muscle enzymes, including: CK, AST, ALT. In particular, CK enzyme has the highest specificity for DM while AST and ALT appearing in other organs is an important indicator of disease activity.

In this study, elevated serum CK enzyme was noted in 28 (38.9%) of the cases as compared to 95% noted by Stonecipher et al. [13], 75% noted by Prellwitz et al. [14], and 65% noted by Gowdie et al [10]. CK was also found to be normal in many instances of DM, which does not appear to be required for diagnosis or related to prognosis. CK can be helpful to aid in the diagnosis but must be interpreted in the context of patient's complete clinical symptoms of proximal myopathy and cutaneous involvement and investigational studies including myopathic pattern on electromyography, muscle and skin biopsy [15].

The presence of autoantibodies in serum is one of the gold standards for the diagnosis of autoimmune diseases. DM is a systemic disease with diverse clinical manifestations and the presence of multiple autoantibodies in the serum. Among 63 patients tested for autoantibodies, MSAs and MAAs were detected with the prevalence of 71.4% and 36.5%, respectively. In another Vietnamese study conducted by Thuy Nguyen Thi Phuong et al., the seropositive prevalence of MSAs was 54% [8].

In our study, the frequency of autoantibodies was as follows: anti-TIF1γ in 18 (28.6%), anti-Ro52 in 14 (22.2%), anti-synthetase in 11 (17.5%), anti-Mi-2 in 9 (14.3%), anti-MDA5 in 9 (14.3%), anti-Ku in 8 (12.7%). Other antibodies were positive with low rate, ranging from 0–7.9%. A study by Ikeda et al. analyzed 55 DM patients and found that six patients (11%) had anti-CADM-140 Ab, nine (16%) had anti-155⁄140 Ab, eight (15%) had ARS-Abs and six (11%) had anti-Mi-2 Ab. The frequency of DM patients positive for any type of autoantibody was 53% [16]. Whereas, Hamaguchi et al. reported that anti-Mi-2, anti-155/140, and anti-CADM-140 were detected in 9 (2%), 25 (7%), and 43 (11%), respectively [4].

In the current study, ARS-Abs was associated with fever, Raynaud's phenomenon and arthralgia. Fukamatsu et al. reported that patients with ARS-Abs more frequently presented with fever and arthralgia, and had elevated levels of C-reactive protein [17]. Hamaguchi et al. also found that myositis was closely associated with anti-Jo-1, anti-EJ, and anti-PL-7, while ILD was correlated with all 6 ARS-Abs. DM-specific skin manifestations (heliotrope rash and Gottron's sign) were frequently observed in patients with anti-Jo-1, anti-EJ, anti-PL-7, and anti-PL-12 [4]. ARS-Abs are an important risk factor for ILD in DM patients. This suggests that we should use HRCT for ILD evaluation for all DM patients.

We found that anti-TIF1γ was associated with V-sign and poikiloderma. Two different subsets were described in patients with anti-TIF1γ: adult malignancy-associated DM and juvenile DM [18, 19]. Although anti-TIF1γ was associated with malignancy, only one DM adult patient (5.6%) had malignancy (nasopharyngeal cancer) in this study. Gunawardena et al. demonstrate that juvenile DM patients with anti-p155/140 autoantibodies had significantly more cutaneous involvement including Gottron's papules (p = 0.003), ulceration (p = 0.005) and oedema (p = 0.013). The distribution of skin lesions was more extensive, particularly peri-orbital (p = 0.014) and over the small (p< 0.001) and large joints (p = 0.003) [20].

Anti-Ro52, anti-Mi-2 and anti-MDA5 are three important antibodies, although our study results showed no statistically significant difference between clinical features and these antibodies. Previous studies found that anti-Mi-2 is associated with typical cutaneous lesions and mild to moderate muscle involvement and responds well to corticosteroid treatment [21, 22]. Anti-Mi-2 portends a benign prognosis and is not associated with an increased risk of the development of malignancy or ILD [23]. 22.2% of our patients had anti-Ro52 antibody, much lower than the results of other studies such as those by Temmoku et al. and Ghillani et al., who found 41.4% and 37%, respectively [24, 25]. Anti-Ro52 has been shown to play a role in the mechanism of many autoimmune diseases such as lupus erythematosus, systemic sclerosis and also associated with ILD [26]. According to Temmoku et al., the presence of anti-Ro52 itself does not affect the phenotype in DM patients. However, when anti-Ro52 is detected in DM patients positive with anti-MDA5 antibody, anti-Ro52 exerts an effect on the clinical course often associated with lower survival rate, increased incidence and severity of ILD in DM [24]. Anti-MDA5 DM is associated with an increased risk of developing ILD, which in some cases may be rapidly progressive (RP-ILD). RP-LD is characterized by short-interval (under 4 weeks) progression of ILD by subjective symptoms or objective metrics (ground glass opacity on computed tomography, worsening PaO2) [27, 28].

This is the first study on autoantibodies and clinical relevance in DM patients in dermatology in Vietnam. A major limitation of this study is the small number of patients analyzed. During the study period, we were able to enroll only 63 autoantibodies-tested DM patients (adults and children) from our hospital. The frequency of these DM-specific autoantibodies is low because each of the autoantibody-based groups has a limited number of patients. Another limitation is this study was conducted in the NHDV, where we are treating skin-related patients with less severe internal organs affected. Severe organ involvements such as rapidly progressive ILD are treated in different centers. Therefore, more studies especially multicenter research

involving a large number of patients are needed for a better general understanding of Vietnamese patients with DM- specific Abs.

## Conclusions

In summary, clinical features in dermatomyositis are heterogeneous. Our study results show some associations between clinical features and autoantibodies in patients with DM. The analysis of DM-related autoantibodies is clinically useful, will be essential for the approaches to diagnosis, and management of DM patients.

## Supporting information

**S1 Data. Dermatomyositis data.**
(SAV)

**S1 File. Dermatomyositis output statistical analysis.**
(PDF)

## Author Contributions

**Conceptualization:** My Huyen Le, Hoa Thi Dinh, Thu Thi Hoai Le, Phuong Thi Hoang, Minh Nguyet Vu, Doanh Huu Le.

**Data curation:** My Huyen Le, Hoa Thi Dinh, Phuong Thi Hoang.

**Formal analysis:** My Huyen Le, Hoa Thi Dinh, Thu Thi Hoai Le.

**Funding acquisition:** My Huyen Le.

**Investigation:** My Huyen Le, Hoa Thi Dinh, Thu Thi Hoai Le, Phuong Thi Hoang, Minh Nguyet Vu.

**Methodology:** My Huyen Le, Hoa Thi Dinh, Thu Thi Hoai Le, Doanh Huu Le.

**Project administration:** My Huyen Le, Phuong Thi Hoang, Minh Nguyet Vu, Doanh Huu Le.

**Resources:** My Huyen Le, Hoa Thi Dinh, Phuong Thi Hoang, Minh Nguyet Vu, Doanh Huu Le.

**Software:** My Huyen Le, Hoa Thi Dinh, Thu Thi Hoai Le.

**Supervision:** My Huyen Le, Phuong Thi Hoang, Minh Nguyet Vu, Doanh Huu Le.

**Validation:** My Huyen Le, Phuong Thi Hoang, Minh Nguyet Vu, Doanh Huu Le.

**Visualization:** My Huyen Le.

**Writing – original draft:** My Huyen Le, Hoa Thi Dinh, Thu Thi Hoai Le.

**Writing – review & editing:** My Huyen Le, Hoa Thi Dinh, Minh Nguyet Vu, Doanh Huu Le.

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
