## [Decision Letter · Decision Letter 0]

14 Jul 2022

PGPH-D-22-00845

THE ASSOCIATION BETWEEN CLINICAL, SUBCLINICAL FEATURES AND AUTOANTIBODY PROFILE IN VIETNAMESE DERMATOMYOSITIS PATIENTS

Dear Le,

Thank you for submitting your manuscript to PLOS Global Public Health. After careful consideration, we feel that it has merit but does not fully meet PLOS Global Public Health’s publication criteria as it currently stands. Therefore, we invite you to submit a revised version of the manuscript that addresses the points raised during the review process.

We look forward to receiving your revised manuscript.

Kind regards,

Collins Otieno Asweto, PhD

Academic Editor

Journal Requirements:

1. Please provide a complete Data Availability Statement in the submission form, ensuring you include all necessary access information or a reason for why you are unable to make your data freely accessible. If your research concerns only data provided within your submission, please write "All data are in the manuscript and/or supporting information files" as your Data Availability Statement.

2. We have noticed that you have uploaded Supporting Information files, but you have not included a list of legends. Please add a full list of legends for your Supporting Information files after the references list. 

Reviewers' comments:

Reviewer's Responses to Questions

**Comments to the Author**

1. Does this manuscript meet PLOS Global Public Health’s publication criteria? Is the manuscript technically sound, and do the data support the conclusions? The manuscript must describe methodologically and ethically rigorous research with conclusions that are appropriately drawn based on the data presented.

Reviewer #1: No

Reviewer #2: Yes

2. Has the statistical analysis been performed appropriately and rigorously?

Reviewer #1: No

Reviewer #2: Yes

3. Have the authors made all data underlying the findings in their manuscript fully available (please refer to the Data Availability Statement at the start of the manuscript PDF file)?

Reviewer #1: No

Reviewer #2: Yes

4. Is the manuscript presented in an intelligible fashion and written in standard English?

Reviewer #1: Yes

Reviewer #2: No

5. Review Comments to the Author

Reviewer #1: 1. the title is vague because the comparison is not clear; is clinical and subclinical feature compared with autoantibody profile?

for me, the comparison seems among the three ( clinical, subclinical, and autoantibody profiles), and the clinical and subclinical features are not stated separately.

2. the study participants, according to the title, are dermatomyositis patients but in the methodology part and in the result the study participants are diabetic patients( 72 DM patients). and again the patients with different diagnoses( dermatitis and myositis) are included in the title. I don't, do the authors mean patients with both manifestations? if that is so, in the metrology the study participants should be patients having both dermatitis and myositis.

3. Almost all parts of the introduction talk about DM which is not part of the study. I knew that DM has many complications on different organs. It can be mentioned that one of the causes of autoantibody profile change will be DM, but in this manuscript, DM was mentioned as the study is about DM.

4. the study participants were DM in the methodology but in the title, it is mentioned that the study participants were dermatomyositis.

5. the statistical analysis was not clearly written. for Example the soft was for data entry was not mentioned, the presentation of variables like categorical and continuous were not explained well.

6. the first paragraph, in the result part, is not in line with the title.

7. the first paragraph of the discussion focused on DM which is not part of the study

8. the way the authors concluded is not appropriate. the result should be about the association between clinical and subclinical futures and autoantibody profile among dermatomyositis patients, and the conclusion will be drawn from the result. but the conclusion was about DM which should never ever be the result of this study.

Reviewer #2: Most of results were interesting to understand and the data is comprehensive and well presented in the tables, and the experimental approaches appear sound. However, the manuscript needs major modifications to be considered for possible publication

First of all, text should be reviewed and corrected by native speaker.

Authors should redescribe Discussion part. The Discussion should not be repeated on the Results. The results of this study should be compared with those of previous studies and the cause of these differences should be described

6. PLOS authors have the option to publish the peer review history of their article (what does this mean?). If published, this will include your full peer review and any attached files.

**Do you want your identity to be public for this peer review?** For information about this choice, including consent withdrawal, please see our Privacy Policy.

Reviewer #1: No

Reviewer #2: No

---

## [Decision Letter · Decision Letter 1]

19 Sep 2022

PGPH-D-22-00845R1

THE ASSOCIATION BETWEEN CLINICAL, SUBCLINICAL FEATURES AND AUTOANTIBODY PROFILE IN VIETNAMESE DERMATOMYOSITIS PATIENTS

Dear Le,

Thank you for submitting your manuscript to PLOS Global Public Health. After careful consideration, we feel that it has merit but does not fully meet PLOS Global Public Health’s publication criteria as it currently stands. Therefore, we invite you to submit a revised version of the manuscript that addresses the points raised during the review process.

We look forward to receiving your revised manuscript.

Kind regards,

Collins Otieno Asweto, PhD

Academic Editor

Journal Requirements:

Reviewers' comments:

Reviewer's Responses to Questions

**Comments to the Author**

1. If the authors have adequately addressed your comments raised in a previous round of review and you feel that this manuscript is now acceptable for publication, you may indicate that here to bypass the “Comments to the Author” section, enter your conflict of interest statement in the “Confidential to Editor” section, and submit your "Accept" recommendation.

Reviewer #2: All comments have been addressed

Reviewer #3: (No Response)

2. Does this manuscript meet PLOS Global Public Health’s publication criteria? Is the manuscript technically sound, and do the data support the conclusions? The manuscript must describe methodologically and ethically rigorous research with conclusions that are appropriately drawn based on the data presented.

Reviewer #2: Yes

Reviewer #3: No

3. Has the statistical analysis been performed appropriately and rigorously?

Reviewer #2: No

Reviewer #3: No

4. Have the authors made all data underlying the findings in their manuscript fully available (please refer to the Data Availability Statement at the start of the manuscript PDF file)?

Reviewer #2: Yes

Reviewer #3: No

5. Is the manuscript presented in an intelligible fashion and written in standard English?

Reviewer #2: Yes

Reviewer #3: No

6. Review Comments to the Author

Reviewer #2: The authors have tried to respond well to major concerns, but still the manuscript requires some improvement

Abstract

Introduction: The authors here they have just provided the knowledge gap or statement of the problem. The authors should provide background information on dermatomyositis and auto-antibodies

Conclusion: The conclusion by the authors is vague, they should provide a concrete conclusion without necessary providing statements on the results

Materials and Methods

The authors used The EUROLINE Immunoblot assay and goes ahead and states that “The incubation protocol includes the following steps:” This sections should always be written in continuous prose instead of stating the steps

Statistical methods

The authors should be more specific how Pearson’s χ2 test or Fisher’s exact test was employed and in respect to which analysis

Discussion section

Conclusion: The conclusion by the authors is vague, they should provide a concrete conclusion without necessary providing statements on the results

Reviewer #3: (No Response)

7. PLOS authors have the option to publish the peer review history of their article (what does this mean?). If published, this will include your full peer review and any attached files.

**Do you want your identity to be public for this peer review?** For information about this choice, including consent withdrawal, please see our Privacy Policy.

Reviewer #2: No

Reviewer #3: No

---

## [Decision Letter · Decision Letter 2]

21 Dec 2022

THE ASSOCIATION BETWEEN CLINICAL, SUBCLINICAL FEATURES AND AUTOANTIBODY PROFILE IN VIETNAMESE DERMATOMYOSITIS PATIENTS

PGPH-D-22-00845R2

Dear Huyen,

We are pleased to inform you that your manuscript 'THE ASSOCIATION BETWEEN CLINICAL, SUBCLINICAL FEATURES AND AUTOANTIBODY PROFILE IN VIETNAMESE DERMATOMYOSITIS PATIENTS' has been provisionally accepted for publication in PLOS Global Public Health.

Best regards,

Collins Otieno Asweto, PhD

Academic Editor

Reviewer Comments (if any, and for reference):

Reviewer's Responses to Questions

**Comments to the Author**

1. If the authors have adequately addressed your comments raised in a previous round of review and you feel that this manuscript is now acceptable for publication, you may indicate that here to bypass the “Comments to the Author” section, enter your conflict of interest statement in the “Confidential to Editor” section, and submit your "Accept" recommendation.

Reviewer #4: (No Response)

Reviewer #5: All comments have been addressed

2. Does this manuscript meet PLOS Global Public Health’s publication criteria? Is the manuscript technically sound, and do the data support the conclusions? The manuscript must describe methodologically and ethically rigorous research with conclusions that are appropriately drawn based on the data presented.

Reviewer #4: Yes

Reviewer #5: Yes

3. Has the statistical analysis been performed appropriately and rigorously?

Reviewer #4: Yes

Reviewer #5: Yes

4. Have the authors made all data underlying the findings in their manuscript fully available (please refer to the Data Availability Statement at the start of the manuscript PDF file)?

Reviewer #4: Yes

Reviewer #5: Yes

5. Is the manuscript presented in an intelligible fashion and written in standard English?

Reviewer #4: No

Reviewer #5: Yes

6. Review Comments to the Author

Reviewer #4: Please consider the following edits in the sections as mentioned below:

Abstract results section:

- Most common signs should be organized from more common to least.

- Aspartate aminotransferase instead of aspartate transanmine

Introduction section

- 2-10/1.000.000 should be replaced with 2-10/1,000,000

- Literature reviews indicates that the presence of different autoantibodies varied across different ethnicities and is associated with clinical features and prognosis of the DM.

- Anti-Jo-1 is one of the anti-synthetase antibodies, which has an important value in the diagnosis of DM. It is also a prognostic factor with over 70% of positive anti-Jo-1 DM patients have reported to develope interstitial lung disease (ILD).

- In contrast to anti-Jo-1, positive anti-Mi-2 is often associated with typical skin lesions but not severe muscle damage and often is known to respond well to treatment

- The proportion of these patients associated with cancer ranges from 42 to 75%

- Thus, anti-MDA5 antibody levels can be used as a biomarker of disease activity and as a predictor of response to treatment

Results section

- Under table 1: p-value: Comparing the gender difference between adult DM and JDM

- Replace Itchy with pruritus; lose weight with weight loss

Discussion:

- According to results from previous studies, DM can occur at any age, but is found to be more common in these two age groups:

- Insert space: Thefemale-male ratio was 1.7

- Other systemic symptoms observed includes include: hair loss 27.8%, fever 12.5%, lymphadenopathy 13.9%, pruritus 5.8%

Reviewer #5: The authors have done a good job addressing concerns raised in prior reviews.

7. PLOS authors have the option to publish the peer review history of their article (what does this mean?). If published, this will include your full peer review and any attached files.

**Do you want your identity to be public for this peer review?** For information about this choice, including consent withdrawal, please see our Privacy Policy.

Reviewer #4: **Yes: **Ronald M. Gobina

Reviewer #5: No
